# Contrasting Roles of Programmed Death-Ligand 1 Expression in Tumor and Stroma in Prognosis of Esophageal Squamous Cell Carcinoma

**DOI:** 10.3390/cancers16061135

**Published:** 2024-03-13

**Authors:** Tomohiro Murakami, Eisuke Booka, Satoru Furuhashi, Yuki Sakai, Kenichi Sekimori, Ryoma Haneda, Mayu Fujihiro, Tomohiro Matsumoto, Yoshifumi Morita, Hirotoshi Kikuchi, Yoshihiro Hiramatsu, Satoshi Baba, Hiroya Takeuchi

**Affiliations:** 1Department of Surgery, Hamamatsu University School of Medicine, 1-20-1 Handayama, Chuo-ku, Hamamatsu City 431-3192, Shizuoka, Japan; tmura626@hama-med.ac.jp (T.M.);; 2Department of Diagnostic Pathology; Hamamatsu University School of Medicine, 1-20-1 Handayama, Chuo-ku, Hamamatsu City 431-3192, Shizuoka, Japan; 3Department of Perioperative Functioning Care and Support, Hamamatsu University School of Medicine, 1-20-1 Handayama, Chuo-ku, Hamamatsu City 431-3192, Shizuoka, Japan

**Keywords:** esophageal squamous cell carcinoma, PD-L1 expression, immunohistochemistry, prognostic factor, chemotherapy response

## Abstract

**Simple Summary:**

Esophageal squamous cell carcinoma (ESCC) often recurs in advanced stages. With immune checkpoint inhibitors becoming a treatment option for recurrent ESCCs, assessing programmed death-ligand 1 (PD-L1) expression in tumor tissue has gained importance. However, challenges remain due to subjective interpretation methods, the unclear significance of staining intensity, and the contrasting roles of stromal and tumoral PD-L1 expression. Our study addresses these issues through a robust, machine learning-aided calculation of H-scores for PD-L1 expression, conducting separate analyses of tumoral and stromal areas in 194 cases of surgically resected ESCCs. Survival analysis revealed that high tumoral PD-L1 expression is an independent factor associated with prolonged survival. In contrast, high stromal PD-L1 expression correlated with less advanced pathological stages and a prolonged response to cytotoxic chemotherapy. This study highlights distinct PD-L1 roles in ESCC’s tumoral and stromal areas, underlining the importance of understanding these mechanisms in the tumor immune microenvironment to develop better treatments and improve patient outcomes.

**Abstract:**

The assessment of programmed death-ligand 1 (PD-L1) expression in esophageal squamous cell carcinoma (ESCC) has become increasingly important with the rise of immune checkpoint inhibitors (ICIs). However, challenges persist, including subjective interpretation and the unclear significance of staining intensity, as well as contrasting roles in tumoral and stromal regions. Our study enhances the understanding of PD-L1 in ESCCs by analyzing its expression in tumors and stroma with H-scores, highlighting its distinct clinicopathological impacts. In a retrospective cohort of 194 ESCC specimens from surgical resection, we quantified PD-L1 expression in tumoral and stromal compartments using H-scores, analyzing whole slide images with digital pathology analysis software. Kaplan–Meier analysis demonstrated that higher PD-L1 expression is significantly associated with improved postoperative overall survival (OS) and recurrence-free survival (RFS) in both tumoral and stromal areas. Multivariable analysis identified high tumoral PD-L1 expression as an independent prognostic factor for prolonged OS and RFS (HR = 0.47, *p* = 0.007; HR = 0.54, *p* = 0.022, respectively). In a separate analysis, high stromal PD-L1 expression was found to correlate with less advanced pathological stages and a prolonged response to cytotoxic chemotherapy, with no similar correlation found for ICI treatment response. This study reveals PD-L1’s contrasting role in the ESCC tumor immune microenvironment, impacting prognosis, tumor stage, and treatment response.

## 1. Introduction

Esophageal squamous cell carcinoma (ESCC) is a major global health concern, ranking as the sixth leading cause of cancer-related deaths worldwide [1]. While adenocarcinoma is more prevalent in Western countries, ESCC is particularly common in Asian countries, including Japan and China, accounting for up to 90% of esophageal cancers in these regions [2]. The standard treatment for stage II and III ESCC varies by region, with neoadjuvant cytotoxic chemotherapy (CTx) followed by surgery being common in Japan, and chemoradiotherapy (CRT) followed by surgery being common in Western countries [3,4,5]. A comprehensive nationwide database in Japan, the National Clinical Database (NCD), highlighted a significant reduction in 90-day mortality rates for esophagectomies, dropping from 3.2% in 2011 to 1.5% in 2020. Concurrently, the use of minimally invasive esophagectomy techniques rose markedly from 31.0% to 71.3%, reflecting a trend toward minimally invasive surgery [6]. Efforts to enhance postoperative outcomes and quality of life continue, focusing on tailoring treatment strategies and reconstructive procedures to individual patient and oncological needs for improved mid- to long-term outcomes [7]. However, recurrence rates remain high for stage II and III ESCCs, posing a significant challenge [8]. Current standard treatments for recurrent ESCC are diverse and evolving. They include CTx, CRT, and local resection. Recently, immune checkpoint inhibitors (ICIs) have been widely accepted as a first-line treatment option [3,4,5]. Studies such as the ATTRACTION-3 trial have demonstrated that ICIs have shown promising results in previously treated advanced ESCC patients, irrespective of tumor programmed death-ligand 1 (PD-L1) expression [9]. On the other hand, the Keynote-181 study suggested that pembrolizumab may offer a potential benefit in patients with high PD-L1 expression [10]. This indicates that the relationship between PD-L1 expression and ICI treatment response could be more complex than has been previously understood. The assessment of PD-L1 expression using combined positive score (CPS), as detailed in ESCC diagnostic guidelines, presents certain challenges [11]. These include the potential for inconsistency in the interpretation of 1+ staining intensity among pathologists, the unclear clinical significance of intensities exceeding 1+, and the lack of clinical implications of differential expression in tumor cells versus the stromal component. Among the various techniques for quantification, the H-score is particularly effective due to its unique scoring system that emphasizes the presence of strongly stained cells. This method enriches the evaluation of biomarker expression by combining a semi-quantitative assessment of staining intensity with the proportion of positively stained cells. Specifically, it assigns a score from 0 (no staining) to 3 (strong staining) based on the intensity, which is then integrated with the percentage of positive cells to compute the H-score, ranging from 0 to 300 [12]. Recent technological advancements have deepened our understanding of the complexity and diversity of the immune context within the tumor microenvironment. It is now recognized that the immune landscape of tumors is not uniform; rather, it comprises different subclasses of immune environments that significantly influence tumor initiation, progression, and response to therapy. The tumor immune microenvironment (TIME) encompasses a dynamic interplay of various cellular and molecular components, including immune cells, cytokines, and other factors that can either promote or inhibit tumor growth and metastasis [13]. The differential expression of PD-L1 in the tumor and stroma is not well understood at the basic research level. It is generally understood that PD-L1 expression can be induced in the tumor microenvironment under certain conditions, such as in the presence of inflammatory factors like interferons (IFNs), tumor necrosis factor-alpha (TNF-α), cell growth factors, hypoxic conditions, and exosomes [14,15]. Diverse findings exist regarding the role of PD-L1 expression in tumoral and stromal regions, with no consensus and limited research specifically in ESCCs [16,17,18,19,20]. This study delves into the clinicopathological significance of PD-L1 expression in ESCC, aiming to comprehensively understand its role in predicting responses to treatment modalities, including ICIs and CTx. Motivated by the ambiguous nature of PD-L1 as a predictive marker for ICI efficacy, our research seeks to broaden the investigation into its impact on disease progression and prognosis. This involves a detailed evaluation of PD-L1’s staining intensity and its variable implications in tumoral versus stromal areas, aiming to add another layer to current evaluation methods, which may overlook critical aspects of stromal expression. By adopting a holistic approach, our goal is to uncover the multifaceted role of PD-L1 in reflecting the status of the TIME, and its viability as a biomarker for customizing treatment strategies in ESCC. This investigation aims to clarify PD-L1’s prognostic value and refine treatment paradigms, contributing to enhanced patient outcomes in the evolving landscape of ESCC treatment. 

## 2. Materials and Methods

### 2.1. Patients and Samples

Between 23 January 2012, and 2 December 2020, our study enrolled 194 patients diagnosed with ESCC of stage pT1 or higher, and these cases were retrospectively reviewed. All of the patients in this study underwent surgical resection at Hamamatsu University School of Medicine. Patient characteristics are detailed in Table 1. The observation period for this study extended until June 2023, with a median follow-up duration of 60 months (IQR 42), excluding deceased cases.

### 2.2. Immunohistochemistry

Formalin-fixed paraffin-embedded blocks from the tumor center were selected based on the standard procedures of the Department of Diagnostic Pathology, focusing on the deepest tumoral regions, as advised by two pathologists. Sections from these blocks were prepared for immunohistochemical analysis. The sections underwent deparaffinization, the inhibition of endogenous peroxidase using 3% H2O2, and antigen retrieval using Dako Target Retrieval Solution, pH 9 (S236784-2). For the primary antibody, we used the research-grade rabbit monoclonal anti-PD-L1 antibody [28-8] (ab205921), which has been documented for use in various cancers, including esophageal carcinoma, in the literature [21,22]. The primary antibody was diluted at a 1:100 ratio and incubated overnight. The application of the subsequent secondary antibody and 3,3′-Diaminobenzidine (DAB) staining were performed using the EnVision Detection Systems (K500711-2, DAKO, Santa Clara, CA, USA). Counterstaining was achieved using Mayer’s hematoxylin solution. The slides were then dehydrated, cleared, and mounted with a cover slip. The pathologists involved in selection contributed to the confirmation of the stained samples.

### 2.3. Image Acquisition and Analysis

Whole slide image files were created by scanning the slides using NanoZoomer^®^-HT/NanoZoomer^®^2.0-HT (Hamamatsu Photonics, Hamamatsu, Japan) at 20× magnification in brightfield mode and saved as NDPi files. These files were analyzed using Qupath v0.4.3 [23]. Regions of interest (ROI) in the tumoral areas were established referencing the hematoxylin- and eosin-stained specimens that had been previously diagnosed by pathologists in the Department of Diagnostic Pathology. The positive cell selection feature was utilized to detect the sum of optical density (OD) for hematoxylin and DAB staining within the slides. The cell size was set within the range of 10–400 μm, with a cell expansion of 5μm. Stromal and tumoral regions were selected as annotations, and machine learning was applied to classify objects in the ROI as tumor or stroma based on their appearance to the researcher. Thresholds for 1+, 2+, and 3+ intensities were established by selecting the mean ± SD values of the compiled OD for stroma + tumor ROI (Appendix A). Using these thresholds (0.100, 0.134, 0.168), H-scores for stroma, tumor, and stroma + tumor were calculated for the 194 slides (Appendix A).

### 2.4. Statistical Analysis

To determine the most significant H-score cutoff affecting postoperative overall survival (OS) and postoperative recurrence-free survival (RFS), the minimum *p*-value method was employed with the logrank function from the “cutoff” package in R 4.3.0 (R Core Team, 2023. R: A language and environment for statistical computing. R Foundation for Statistical Computing, Vienna, Austria. URL https://www.R-project.org/ (accessed on 9 July 2023) via RStudio Version 2023.06.1+524 (RStudio Team, 2023) [24]. Survival analysis was performed for OS and RFS using the Kaplan–Meier method and analyzed using the logrank test. Cox regression analysis was performed using various clinicopathological factors, including age, gender, differentiation status, pT, pN, lymphatic invasion, venous invasion, intramural metastasis status, and the H-score status of PD-L1 in the stroma, tumor, and stroma + tumor. In the context of this analysis for OS, the PD-L1 H-score cutoff values for stroma, tumor, and stroma + tumor were set based on the value yielding the minimum *p*-value specifically for OS, thereby categorizing patients into two distinct groups. Separately, a univariate analysis was performed, setting the significance level at *p* < 0.05. Variables of clinical significance identified from the univariate analysis were then subjected to a multivariate analysis. For the RFS analysis, an identical approach was adopted, with the only difference using H-score cutoff values that were set based on the minimum *p*-value for RFS. Survival analyses were performed using SPSS Version 29 (IBM Corp., Armonk, NY, USA). Correlation analysis was conducted to assess the relationship between clinicopathological factors and PD-L1 expression status in the stroma, tumor, and stroma + tumor regions. The cutoff values for categorizing PD-L1 expression as high or low were determined based on the H-scores that yielded the minimum *p*-value in OS analysis. Chi-square test analysis was performed using SPSS Version 29 (IBM Corp., Armonk, NY, USA). Statistical significance was set at *p* < 0.05. The H-scores for PD-L1 expression in stroma, tumor, and stroma + tumor were compared using the Kruskal–Wallis and Dunn’s multiple comparison test. For the analysis of treatment duration and PD-L1 H-score, the Mann–Whitney U test was utilized. These analyses were conducted using GraphPad Prism (version 9.0, GraphPad Software, LLC., San Diego, CA, USA).

## 3. Results

### 3.1. PD-L1 Expression in Stroma, Tumor, and Stroma + Tumor

To gain an overview of the distribution of PD-L1 expression in tumor and stroma, we analyzed the median H-scores in our cohort of 194 cases. The results revealed that median H-scores for PD-L1 expression in the tumor were significantly higher than those in the stroma (median 155.0 vs. 97.4; *p* < 0.001, Figure 1A). Representative images of PD-L1 staining using the quartile H-scores of the tumor and stroma as cutoff values are shown (Figure 1B).

### 3.2. Survival Analysis Based on PD-L1 H-Score

To evaluate the clinical significance of PD-L1 expression in tumor, stroma, and stroma + tumor on postoperative prognosis, we performed a survival analysis using PD-L1 H-Scores. Kaplan–Meier survival curves generated using quartile H-scores as cutoff values for stroma, tumor, and stroma + tumor showed a trend between higher H-score and improved prognosis (Appendix A). Next, we attempt to determine the optimal H-Score cutoff values that will give the most clinical impact in the analyzed cases. The minimum *p*-value method for the logrank test was used to identify the optimal cutoff. The identified optimal cutoff values for OS for stroma, tumor, and stroma + tumor were 114.86 (*p* = 0.000020), 159.83 (*p* = 0.002498), and 165.22 (*p* = 0.003303), respectively (Appendix A). The optimal cutoff values for RFS were 114.86 (*p* = 0.000080), 164.22 (*p* = 0.000556), and 185.45 (*p* = 0.004172), respectively (Appendix A). Kaplan–Meier survival curves were generated using the optimal H-score cutoff values. For OS, groups with high H-scores in stroma, tumor, and stroma + tumor showed significantly longer survival (*p* < 0.001, *p* = 0.005, and *p* = 0.007 respectively; Figure 1C–E). Similarly, for RFS, longer survival durations were observed in groups with high H-scores in stroma, tumor, and stroma + tumor (*p* < 0.001, *p* = 0.008, and *p* = 0.009 respectively; Figure 1F–H).

### 3.3. Multivariate Analysis of Clinicopathological Factors

To determine the significance of PD-L1 expression alongside other clinicopathological factors in influencing OS in ESCC, we performed a Cox regression analysis. In the univariate analysis to identify clinicopathological factors affecting OS, preoperative treatment, pT2 or higher, pN positive, lymphatic invasion positive, venous invasion positive, intramural metastasis positive, and low PD-L1 expression in stroma, tumor, and stroma + tumor were identified as significant factors for shorter OS (Table 2). In the multivariate analysis, pN positive (HR = 2.70, 95% CI 1.12–6.52, *p* = 0.027) and low PD-L1 expression in the tumor (HR = 0.47, 95% CI 0.27–0.83, *p* = 0.010) remained significant to predict poor OS. Next, Cox regression analysis was performed to identify clinicopathological factors affecting RFS (Table 3). In the univariate analysis, the inclusion of a poorly differentiated tumor, preoperative treatment, pT2 or higher, pN positive, lymphatic invasion positive, venous invasion positive, intramural metastasis positive, low PD-L1 expression in stroma, tumor, and stroma + tumor were identified as significant factors for shorter RFS. In the multivariate analysis, pN positive (HR = 2.56, 95% C.I. 1.17–5.58, *p* = 0.018) and low PD-L1 expression in the tumor (HR = 0.54, 95%C.I. 0.32–0.91, *p* = 0.022) remained significant to predict poor RFS.

### 3.4. Correlation Analysis of PD-L1 Expression and Clinicopathological Factors

Given that only tumoral PD-L1 expression was identified as a significant factor in the multivariate analysis for survival, we investigated whether specific clinicopathological factors differentially influence PD-L1 expression in the stroma, tumor, and stroma + tumor compartments. Clinicopathological factors correlated with PD-L1 expression in stroma, tumor, and stroma + tumor were analyzed (Table 4). Higher pT Stage, higher pN Stage, positive lymphatic invasion, and positive venous invasion were associated with lower stroma PD-L1(*p* < 0.001, respectively). However, no such correlation was observed for the tumor. For stroma + tumor, significant correlations were observed with pN stage (*p* = 0.041) and venous invasion (*p* = 0.002).

### 3.5. PD-L1 Expression and Its Association with Recurrence Patterns and Metastatic Sites

To further investigate the role of PD-L1 expression in disease progression of ESCC, we conducted an analysis focused on recurrence patterns, total recurrence types, and total metastatic sites in relation to PD-L1 expression in the stroma, tumor, and stroma + tumor regions. This analysis was specifically performed in cases of recurrence (Table 5). Our findings indicate that there were no statistically significant differences across these variables in relation to PD-L1 expression.

### 3.6. Comparison of Survival Times Based on PD-L1 Expression Patterns in Stroma and Tumor

Given that our previous analyses suggested the presence of a PD-L1 expression pattern with distinct prognostic implications, we sought to assess the impact of various combinations of PD-L1 expression in stromal and tumoral regions. These were characterized as either high or low in both compartments, leading us to conduct a comparative analysis of OS and RFS. The minimum *p*-value cutoff for OS and RFS was utilized to categorize PD-L1 expression in stroma and tumor into four bins, resulting in distinct patterns of expression (Figure 2A). When we generated Kaplan-Meier survival curves for these four groups, we found a significant difference in both OS and RFS between the group with both low stromal and tumoral PD-L1 expression and the group with high expression in both regions (*p* < 0.001, respectively Figure 2B,C).

### 3.7. PD-L1 Expression and Post-Recurrence Treatment Efficacy: CTx and Immune Checkpoint Inhibitors

Our observations of varying recurrence rates linked to PD-L1 expression levels led us to examine their potential impact on post-recurrence treatment outcomes, focusing on the duration of CTx and responses to ICIs without progression. Among the 85 cases that recurred, 70 received CTx for recurrence treatment. Our analysis indicated a significantly higher stromal H-score in cases with a CTx treatment period of 8 months or more (median 53.8 vs. 98.3, *p* = 0.028, Figure 2D). However, no significant difference in PD-L1 expression was observed for the tumor (median 133,8 vs. 154.6, *p* = 0.200, Figure 2E) and stroma + tumor (median 113.7 vs. 122.3, *p* = 0.355, Figure 2F). Furthermore, among the recurrent cases, 24 received ICIs. Analysis showed no significant difference in stroma, tumor, or stroma + tumor PD-L1 expression between cases that were able to continue ICIs without progression for 90 days or more and those that did not (median 102.4 vs. 106.6, *p* = 0.450; median 148.5 vs. 158.7, *p* = 0.568; median 108.1 vs. 122.3, *p* = 0.287, respectively; Figure 2G–I).

## 4. Discussion

This study quantified PD-L1 expression in 194 surgical specimens of ESCC using digital pathology, classifying the whole tumor within the slide into tumoral and stromal compartments enhanced using AI. By employing Kaplan–Meier survival analysis and categorizing H-scores into quartiles, we found that higher scores were associated with better OS and RFS. Utilizing the minimum *p* value method for logrank test, we determined statistically optimal H-score cutoff values for OS and RFS, dividing the cohort into groups to assess the impact on survival. In the multivariate analysis regarding survival, tumoral PD-L1 expression emerged as an independent predictor of prognosis for both OS and RFS. Our correlation analysis with clinicopathological variables revealed that high stromal PD-L1 expression was strongly linked to less advanced pathological stages. Furthermore, when comparing H-scores with the duration of CTx and ICI treatment, we discovered that cases treated for more than 8 months exhibited higher stromal H-scores. However, no difference in H-scores between compartments was observed in cases treated with ICI for more than 90 days versus those treated for shorter periods. Our approach to evaluating PD-L1 expression in this study marks a significant shift from traditional methods by leveraging digital pathology for quantitative analysis of PD-L1 in ESCC, thus deepening our grasp of its clinical-pathological significance. Enhanced by AI-based tissue classification, this method efficiently differentiates between tumoral and stromal regions, spotlighting the underexplored area of stromal PD-L1 expression and its relative clinical-pathological importance in comparison to tumoral PD-L1 expression in ESCC. This pioneering approach not only improves the precision of PD-L1 measurement but also expands our insight into its role within the TIME, thereby offering a more nuanced understanding of the interplay between tumoral and stromal components in ESCC. By providing a detailed analysis of PD-L1 expression in both tumoral and stromal compartments, our study contributes to resolving the conflicting perspectives in the broader context of ESCC research, where previous studies have yielded mixed results regarding the prognostic significance of PD-L1 expression. Certain investigations indicated that higher PD-L1 expression in tumor cells was linked to shorter disease-free survival and OS [25]. In contrast, there are studies that found higher PD-L1 expression to be associated with better prognosis [26,27]. The distinguishing of PD-L1 expression in tumoral and stromal compartments is seldom addressed. Some studies have evaluated PD-L1 expression in both compartments but not in ESCC [16,17,18]. Previous research on stromal PD-L1 expression has primarily focused on immune cells within the tumor tissue. Some studies have associated high stromal PD-L1 expression with worse prognosis [17,28] while others have linked it to better prognosis [16,19,20]. In our study, we could not differentiate the stromal expression of PD-L1 in immune cells from other stromal cells upon the classification procedure due to the complexity and heterogeneity of stromal components. It is anticipated that future advancements in image analysis and machine learning will enable the differentiation of subtle features in the TIME, thereby facilitating a better understanding of PD-L1 dynamics in ESCC. Although, this brings us to the intriguing possibility that TIME components such as cancer associated fibroblasts (CAFs), could also have an influence on PD-L1 expression. With regard to PD-L1 expression on CAFs, there are only a handful of studies, and their findings remain inconclusive. One study in nonmetastatic non-small cell lung cancer suggested that PD-L1 expression on CAFs was associated with a better prognosis [20], whereas another study in colorectal cancer reported that CAFs upregulate PD-L1 expression in tumor cells, contributing to worse prognosis [29]. The molecular mechanisms underlying PD-L1 expression in CAFs remain largely unexplored to date. Thus, further investigations are necessary to elucidate the role of CAFs in PD-L1 expression and to determine its prognostic significance. Our study revealed that elevated stromal PD-L1 expression was linked to a longer treatment period (8 months or more) in patients undergoing CTx, implying that stromal PD-L1 expression might serve as a predictive biomarker for a favorable response to CTx. This echoes a study on triple-negative breast cancer, which demonstrated the potential of PD-L1 as a response biomarker for CTx [30]. Our research offers an innovative perspective by examining PD-L1’s influence on CTx effectiveness in ESCC, moving beyond the conventional focus on its correlation with ICI responsiveness. By understanding PD-L1 status from resected specimens, we can predict the responsiveness to CTx, which plays a significant role from the first line recurrence treatment, paving the way for more precise and personalized medicine. As discussed so far, the role of PD-L1 in either promoting progression or exerting a suppressive effect on tumors remains controversial in the literature. Our findings, which indicate that high tumoral PD-L1 expression is associated with better prognosis, align with some previous studies in ESCC. Additionally, our report that higher stromal PD-L1 expression is linked to less advanced disease progression and our discovery that PD-L1 expression influences responsiveness to CTx in ESCC both represent first time in the context of ESCC. However, contrary to the KEYNOTE181 trials, our study did not observe a correlation between PD-L1 expression and an improved response to immunotherapy. This discrepancy could potentially be attributed to the smaller sample size in our study compared to these large clinical trials. It is also possible that other factors, such as variations in treatment protocols or patient characteristics, could have influenced this outcome. Thus, while our findings contribute to the evolving understanding of PD-L1’s role in immunotherapy, more comprehensive studies are needed to definitively ascertain this relationship. PD-L1 expression in cells are known to be influenced by factors like inflammatory cytokines, hypoxic conditions, and exosomes [14,15]. TIME components and the altered cytokine milieu could further contribute to this complexity [31,32]. These factors might be involved in mechanisms that render tumors more responsive to CTx and potentially immunotherapy. Further research is necessary to unravel these complex dynamics and could lead to strategies for effectively targeting stromal PD-L1 to enhance the response to therapy. Building upon our foundational work in delineating PD-L1 expression dynamics within the stromal and tumor cells of ESCC, future research will delve further into the TIME. This exploration includes assessing T-cell and tumor-associated macrophage markers, along with inflammatory cytokines that potentially exacerbate ESCC prognosis through mechanisms related to cachexia. Each step forward in deciphering the intricate interactions within the TIME not only aids in predicting treatment responses but also strives to identify novel therapeutic targets, moving us closer to improving patient outcomes in ESCC. There are several methodological limitations in this study. Future directions will focus on expanding the study scale and diversity by undertaking a multi-institutional approach to compile a larger, more diverse dataset. This expansion should enhance the persuasiveness, generalizability, and credibility of the results by incorporating varied patient demographics and treatment histories. Insights into PD-L1 expression in patients treated with ICIs as a first-line treatment can offer new perspectives on mechanisms of resistance and significantly contribute to our understanding of PD-L1′s role in ESCCs. Integration of multi-omics data from public databases and existing resources for genomic, transcriptomic, and proteomic analyses can deepen our understanding of PD-L1 dynamics and interactions within TIME. Longitudinal analysis will allow us to monitor changes in PD-L1 levels over time, for example, by using endoscopic biopsy specimens before and after treatment could provide insights into PD-L1′s prognostic significance, inform treatment planning, and possibly reveal aspects of treatment responsiveness. Regarding digital pathology analysis, expanding the dataset to include slides evaluated with CPS and TPS will enable a comparison between traditional and digital pathology methods, potentially validating the clinical utility of digital pathology. To sum, this research advances the field by elucidating the complex role of PD-L1 in both tumoral and stromal compartments of ESCC, employing quantification methods to capture staining intensity nuances not considered by CPS and TPS. The digital pathology-based H-score quantification methods offer a more nuanced view of PD-L1’s impact on cancer progression and therapeutic avenues. This work underscores the necessity of a holistic view of PD-L1 expression within the TIME, contributing significantly to the understanding of its role in ESCC and guiding future therapeutic strategies.

## 5. Conclusions

Our study demonstrates that high tumoral PD-L1 expression is an independent prognostic factor for longer survival. Conversely, high stromal PD-L1 expression correlates with less advanced pT and pN stages, negative lymphatic and venous invasion, and prolonged duration of CTx treatment. Contrary to expectations, PD-L1 expression in the stromal, tumor, and stroma + tumor compartments did not affect the duration of ICI treatment.

## Figures and Tables

**Figure 1 cancers-16-01135-f001:**
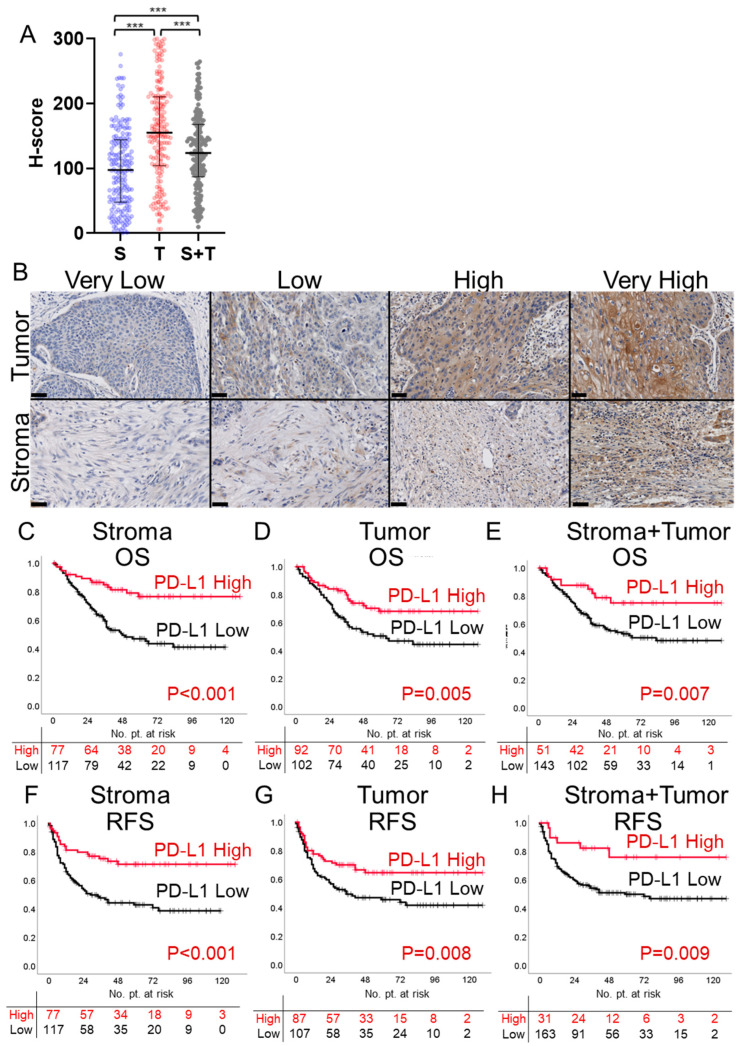
PD-L1 expression and its impact on survival in ESCC patients. (**A**) Scatter bar graph showing the median H-scores for PD-L1 expression in the stroma (S), tumor (T), and stroma + tumor (S + T) in a cohort of 194 ESCC cases. Significant differences among the groups were observed, as determined by the Kruskal–Wallis and Dunn’s multiple comparison test (*** *p* < 0.001). (**B**) Representative PD-L1 staining images selected based on quartile H-scores for tumor and stroma. Scale bar: 50 μm. (**C**–**H**) Kaplan–Meier survival curves illustrating OS and RFS stratified by optimal PD-L1 H-score cutoffs in stroma, tumor, and stroma + tumor. Survival differences between groups were analyzed using the logrank test.

**Figure 2 cancers-16-01135-f002:**
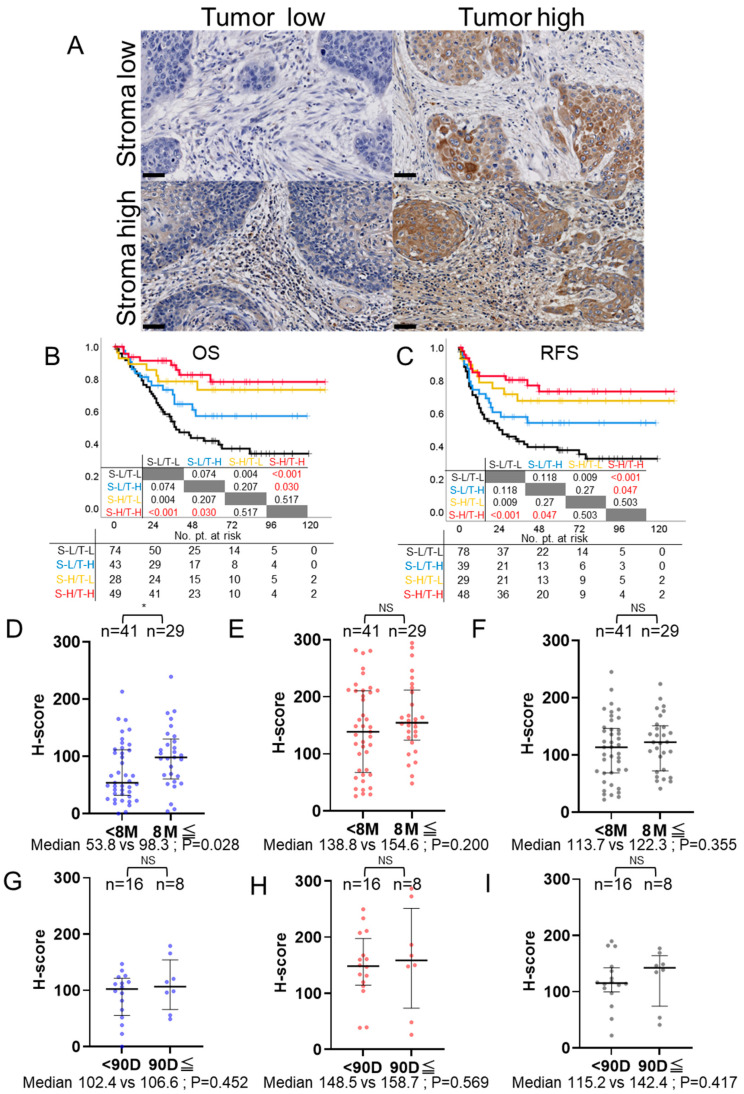
Impact of PD-L1 expression patterns in stroma and tumor on survival times and post-recurrence treatment efficacy. (**A**) Representative image showing four distinct PD-L1 expression patterns in stroma and tumor, categorized based on the minimum *p*-value cutoffs for OS. Scale bar: 50 μm. (**B**,**C**) Kaplan–Meier survival curves comparing OS and RFS among four groups with distinct PD-L1 expression patterns in stroma and tumor, highlighting the impact of PD-L1 expression on survival times. Groups were categorized using minimum *p*-value cutoffs for OS and RFS, respectively. The patterns are abbreviated as follows: stroma low/tumor low (S-L/T-L), stroma low/tumor high (S-L/T-H), stroma high/tumor low (S-H/T-L), and stroma high/tumor high (S-H/T-H). Survival differences between groups were analyzed using the logrank test. (**D**–**I**) Scatter bar graphs illustrating H-scores for PD-L1 expression in the stroma, tumor, and stroma + tumor compartments, focusing on the correlation with post-recurrence treatment efficacy in CTx and ICIs (* *p* < 0.05, NS -not significant). (**D**–**F**) Graphs compare cases that experienced recurrence and received CTx for 8 months or longer versus those treated for less than 8 months. (**G**–**I**) Graphs compare cases that experienced recurrence and received ICIs for 90 days or longer versus those treated for less than 90 days. Median and interquartile ranges are indicated on the graphs. The Mann–Whitney U test was used to assess statistical significance for both comparisons.

**Table 1 cancers-16-01135-t001:** Detailed characteristics of the 194 patients diagnosed with ESCC of stage pT1 or higher analyzed in this study.

Variables	N	(%)
Age		
under 70 y.o	114	(59%)
over and 70 y.o	80	(41%)
Sex		
male	169	(88%)
female	25	(13%)
Tumor Location		
Ce	4	(2%)
Ut	22	(11%)
Mt	95	(49%)
Lt	72	(37%)
Ae	1	(1%)
Differentiation		
differentiated	140	(73%)
include poorly differentiated	41	(21%)
NA	13	(7%)
Preoperative treatment		
None	97	(50%)
Chemotherapy	87	(45%)
Chemotherapy+Radiation	10	(5%)
pT		
T1	94	(49%)
T2	26	(13%)
T3	74	(38%)
pN		
N0	71	(37%)
N1	65	(34%)
N2	34	(18%)
N3	24	(12%)
Lymphatic invasion		
negative	70	(36%)
positive	124	(64%)
Venous invasion		
negative	55	(28%)
positive	139	(72%)
Intramural metastasis		
negative	176	(91%)
positive	18	(9%)
pStage *		
I	55	(28%)
II	43	(22%)
III	59	(31%)
IVA	37	(19%)
Total recurrence cases	85	(44%)
Lymphogenous metastasis		
negative	16	(19%)
positive	69	(81%)
Hematogenous metastasis		
negative	33	(39%)
positive	52	(61%)
Locoregional recurrence		
negative	76	(89%)
positive	9	(11%)
Pleural/peritoneal dissemination		
negative	66	(78%)
positive	19	(22%)

* TNM-8—TNM Classification of Malignant Tumours—8th edition; Ce—Cervical part of the esophagus; Ut—Upper thoracic part of the esophagus; Mt—Middle thoracic part of the esophagus; Lt—Lower thoracic part of the esophagus; Ae—Abdominal part of the esophagus.

**Table 2 cancers-16-01135-t002:** Results of the Cox regression analysis identifying clinicopathological factors affecting OS.

Variables	N	Univariable				Multivariable			
		HR	95%CI		*p*-Value	HR	95%CI		*p*-Value
Age									
under 70 y.o	114	1.00	0.48	1.30	0.355				
over and 70 y.o	80	0.79							
Sex									
male	169	1.00	0.31	1.47	0.321				
female	25	0.67							
Differentiation									
differentiated	140	1.00	0.95	2.79	0.077				
include poorly diff.	41	1.63							
Preoperative treatment									
none	97	1.00	1.77	4.97	<0.001	1.00	0.95	3.08	0.075
chemotherapy(+/-Radiation)	97	2.97				1.71			
pT *									
<T2	94	1.00	2.54	7.82	<0.001	1.00	0.95	3.65	0.069
T2 and over	100	4.46				1.86			
pN *									
negative	71	1.00	2.80	12.28	<0.001	1.00	1.12	6.52	0.027
positive	123	5.86				2.70			
Lymphatic invasion									
negative	70	1.00	1.77	6.11	<0.001	1.00	0.47	2.27	0.943
positive	124	3.28				1.03			
Venous invasion									
negative	55	1.00	1.96	9.35	<0.001	1.00	0.61	3.89	0.357
positive	139	4.23				1.55			
Intramural metastasis									
negative	176	1.00	1.58	5.28	<0.001	1.00	0.80	2.93	0.201
positive	18	2.88				1.53			
PD-L1 Stroma									
low	117	1.00	0.17	0.55	<0.001	1.00	0.21	1.17	0.108
high	77	0.30				0.49			
PD-L1 Tumor									
low	102	1.00	0.30	0.82	0.006	1.00	0.27	0.83	0.010
high	92	0.49				0.47			
PD-L1 Stroma+Tumor									
low	143	1.00	0.20	0.80	0.010	1.00	0.45	3.84	0.616
high	51	0.40				1.32			

* TNM-8—TNM Classification of Malignant Tumours—8th edition.

**Table 3 cancers-16-01135-t003:** Results of the Cox regression analysis identifying clinicopathological factors affecting RFS.

Variables	N	Univariable				Multivariable			
		HR	95%CI		*p*-Value	HR	95%CI		*p*-Value
Age									
under 70 y.o	114	1.00	0.52	1.26	0.344				
over and 70 y.o	80	0.81							
Sex									
male	169	1.00	0.25	1.18	0.121				
female	25	0.54							
Differentiation									
differentiated	140	1.00	1.05	2.74	0.031	1.00	0.71	1.92	0.534
include poorly diff.	41	1.70				1.17			
Preoperative treatment									
none	97	1.00	1.91	4.79	<0.001	1.00	0.99	2.98	0.051
chemotherapy(+/−Radiation)	97	3.03				1.72			
pT *									
<T2	94	1.00	2.36	6.13	<0.001	1.00	0.89	2.91	0.114
T2 and over	100	3.80				1.61			
pN *									
negative	71	1.00	2.68	9.15	<0.001	1.00	1.17	5.58	0.018
positive	123	4.96				2.56			
Lymphatic invasion									
negative	70	1.00	1.66	4.82	<0.001	1.00	0.53	2.15	0.850
positive	124	2.83				1.07			
Venous invasion									
negative	55	1.00	1.71	5.80	<0.001	1.00	0.62	3.03	0.436
positive	139	3.14				1.37			
Intramural metastasis									
negative	176	1.00	1.44	4.56	0.001	1.00	0.70	2.44	0.397
positive	18	2.56				1.31			
PD-L1 Stroma									
low	117	1.00	0.24	0.65	<0.001	1.00	0.43	1.48	0.465
high	77	0.39				0.79			
PD-L1 Tumor									
low	108	1.00	0.35	0.87	0.010	1.00	0.32	0.91	0.022
high	87	0.55				0.54			
PD-L1 Stroma+Tumor									
low	163	1.00	0.15	0.80	0.013	1.00	0.33	2.77	0.930
high	31	0.35				0.95			

* TNM-8—TNM Classification of Malignant Tumours.

**Table 4 cancers-16-01135-t004:** Correlation analysis between PD-L1 expression in stroma, tumor, and stroma + tumor, with various clinicopathological factors using the chi-square test.

Variables	PD-L1 Stroma			PD-L1 Tumor			PD-L1 Stroma + Tumor		
	Low N = 117	High N = 77	*p*-Value	Low N = 102	High N = 92	*p*-Value	Low N = 143	High N = 51	*p*-Value
Age									
under 70 y.o	37%	22%	0.333	31%	27%	0.756	44%	14%	0.514
over and 70 y.o	23%	18%		21%	20%		29%	12%	
Sex									
male	55%	32%	0.074	47%	40%	0.357	65%	22%	0.237
female	6%	7%		6%	7%		8%	5%	
Tumor Location									
Ce	2%	1%	0.736	2%	1%	0.298	2%	1%	0.498
Ut	7%	4%		7%	5%		8%	4%	
Mt	29%	20%		28%	21%		37%	12%	
Lt	22%	15%		16%	21%		27%	10%	
Ae	0%	1%		0%	1%		0%	1%	
Differentiation									
differentiated	42%	30%	0.701	36%	37%	0.267	53%	19%	0.962
include poorly differentiated	14%	7%		12%	9%		15%	6%	
NA	4%	3%		5%	2%		5%	2%	
Preoperative treatment									
None	28%	22%	0.394	28%	22%	0.538	35%	15%	0.510
Chemotherapy	29%	16%		22%	23%		35%	10%	
Chemotherapy+Radiation	4%	2%		3%	2%		4%	1%	
pT *									
T1	22%	27%	<0.001	25%	23%	0.988	32%	16%	0.051
T2	10%	4%		7%	6%		10%	3%	
T3	29%	9%		20%	18%		31%	7%	
pN *									
N0	15%	21%	<0.001	18%	19%	0.362	23%	14%	0.041
N1	22%	11%		21%	13%		26%	7%	
N2	13%	5%		8%	9%		14%	3%	
N3	10%	3%		6%	6%		10%	2%	
Lymphatic invasion									
negative	16%	20%	<0.001	19%	18%	0.810	24%	12%	0.057
positive	44%	20%		34%	30%		50%	14%	
Venous invasion									
negative	10%	19%	<0.001	13%	15%	0.352	16%	12%	0.002
positive	51%	21%		39%	32%		57%	14%	
Intramural metastasis									
negative	54%	37%	0.278	47%	43%	0.790	67%	24%	0.880
positive	7%	3%		5%	4%		7%	3%	
pStage *									
I	8%	20%	<0.001	14%	14%	0.658	16%	12%	0.013
II	17%	5%		11%	11%		17%	5%	
III	20%	10%		18%	12%		24%	7%	
IVA	15%	4%		10%	9%		16%	3%	

* TNM-8—TNM Classification of Malignant Tumours—8th edition; Ce—Cervical part of the esophagus; Ut—Upper thoracic part of the esophagus; Mt—Middle thoracic part of the esophagus; Lt—Lower thoracic part of the esophagus; Ae—Abdominal part of the esophagus.

**Table 5 cancers-16-01135-t005:** Correlation analysis of recurrence patterns, total recurrence types, and total metastatic sites with PD-L1 expression in stroma, tumor, and stroma + tumor regions using the chi-square test.

	PD-L1 Stroma			PD-L1 Tumor			PD-L1 Stroma + Tumor		
Variables	Low N = 65	High N = 20	*p*-Value	Low N = 52	High N = 33	*p*-Value	Low N = 79	High N = 6	*p*-Value
Lymphogenous metastasis									
negative	22%	10%	0.248	23%	12%	0.339	20%	0%	0.135
positive	78%	90%		77%	88%		80%	100%	
Hematogenous metastasis									
negative	35%	50%	0.241	40%	36%	0.247	38%	50%	0.127
positive	65%	50%		60%	64%		62%	50%	
Locoregional recurrence									
negative	86%	100%	0.078	88%	91%	0.386	89%	100%	0.289
positive	14%	0%		12%	9%		11%	0%	
Pleural/peritoneal dissemination									
negative	78%	75%	0.745	79%	76%	0.739	78%	67%	0.503
positive	22%	25%		21%	24%		22%	33%	
Total recurrence types									
1	45%	55%	0.515	46%	48%	0.139	43%	50%	0.742
2	37%	25%		38%	27%		32%	17%	
3	14%	20%		15%	15%		13%	33%	
4	5%	0%		0%	9%		3%	0%	
Total metastatic sites									
1	40%	55%	0.352	44%	42%	0.318	41%	50%	0.115
2	22%	10%		25%	9%		19%	0%	
3	23%	20%		19%	27%		19%	17%	
4	9%	5%		8%	9%		8%	17%	
5	6%	5%		4%	9%		4%	17%	
6	0%	5%		0%	3%		0%	0%	

## Data Availability

The data that support the findings of this study are available from the corresponding author, [T.M.], upon reasonable request.

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
