# Peer review of "Contrasting Roles of Programmed Death-Ligand 1 Expression in Tumor and Stroma in Prognosis of Esophageal Squamous Cell Carcinoma"

_cancers, 2024, doi:10.3390/cancers16061135_

Round 1

Reviewer 1 Report

Comments and Suggestions for Authors

The authors have used a special variant of machine learning to classify region
of interest (ROI) for the expression of PD-L1 within stroma and tumor regions in
Esophageal Squamous Cell Carcinoma. To use AI technique to retrospectively
review 194 ESCC patient samples and the clinicopathological impact of PD-L1
expression with varying implications on disease progression is of novel
perspective and of great interest. The paper is nice to read and the authors
present their study in a pedagogical way and also highlight the obstacles and
ideas of how to continue.
However, there are some questions raised that need to be clarified:
- The analysis is based on formalin-fixed paraffin embedded samples and
selected on pathological reports. How many pathologists were involved
in the selection and confirmation of stained samples in this procedure?
- The use of this specific anti-PD-L1 antibody (ab205921), is this based on
previous testing of anti-PD-L1 antibodies? In some cases, as for the
detection of PD-L1 expression in lung cancer, the used anti-PD-L1 clone is
of great importance for the outcome in the analysis of the cancer
samples.
- “in our study, we could not differentiate the stromal expression of PD-L1
in immune cells from other stromal cells upon the classification
procedure…” what is the potential obstacle regarding this for your study
outcome. Please, reflect on this.
- This study focuses on the PD-L1 status in the tumor, stroma and
combined stroma + tumor. Have you also looked at other possible
candidates as biomarkers for prognosis of ESCC?
Major revision points:
o Figure 1 and figure 2 need to be increased size wise, since the
image quality and size of images, letters and numbers make it hard
to evaluate.
o Table legends are placed below the tables, but should be placed
above.
o Table legend for Table 4 is too scarce, statistical method used?
o Table legend for Table 5 is too scarce, statistical method used?
Minor revision points
o Increase the size of the images for Figure S1, as well
o Look over the heterogeneity how the statistics is presented in the text
section

Author Response

March 7, 2024

Dear Reviewer 1,

We are deeply grateful for your constructive feedback on our manuscript, "Contrasting Roles of PD-L1 Expression in Tumor and Stroma in Prognosis of Esophageal Squamous Cell Carcinoma." Your insights have been instrumental in guiding the revisions that have significantly enhanced the clarity, depth, and impact of our work.

In response to your comments, we have made several revisions to address the concerns raised, including clarifying our methodology, and future directions of the research. Additionally, we have made adjustments to improve the presentation and readability of figures and tables and statistics as suggested.

We believe these changes have strengthened the manuscript and hope that it now meets the high standards of "Cancers" We are eager to hear your thoughts on the revised version and look forward to the possibility of our study being published in this esteemed journal.

Thank you once again for your invaluable feedback and guidance.

Sincerely,

Eisuke Booka, MD, PhD

Second Department of Surgery, Hamamatsu University School of Medicine

1-20-1 Handayama, Chuo-ku, Hamamatsu City, Shizuoka, 431-3192, Japan

Phone: +81-53-435-2279

Response to Reviewer 1 Comments

1. Summary

Thank you very much for taking the time to review this manuscript. Please find the detailed responses below and the corresponding revisions highlighted in the re-submitted files.

2. Questions for General Evaluation

Reviewer’s Evaluation

Response and Revisions

Does the introduction provide sufficient background and include all relevant references?

Yes/Can be improved/Must be improved/Not applicable

Are all the cited references relevant to the research?

Yes/Can be improved/Must be improved/Not applicable

Is the research design appropriate?

Yes/Can be improved/Must be improved/Not applicable

Are the methods adequately described?

Yes/Can be improved/Must be improved/Not applicable

Are the results clearly presented?

Yes/Can be improved/Must be improved/Not applicable

Are the conclusions supported by the results?

Yes/Can be improved/Must be improved/Not applicable

3. Point-by-point response to Comments and Suggestions for Authors

Comments: The authors have used a special variant of machine learning to classify region of interest (ROI) for the expression of PD-L1 within stroma and tumor regions in Esophageal Squamous Cell Carcinoma. To use AI technique to retrospectively review 194 ESCC patient samples and the clinicopathological impact of PD-L1 expression with varying implications on disease progression is of novel perspective and of great interest. The paper is nice to read and the authors present their study in a pedagogical way and also highlight the obstacles and ideas of how to continue. However, there are some questions raised that need to be clarified:

-The analysis is based on formalin-fixed paraffin embedded samples and selected on pathological reports. How many pathologists were involved in the selection and confirmation of stained samples in this procedure?

- The use of this specific anti-PD-L1 antibody (ab205921), is this based on previous testing of anti-PD-L1 antibodies? In some cases, as for the detection of PD-L1 expression in lung cancer, the used anti-PD-L1 clone is of great importance for the outcome in the analysis of the cancer samples.

- “in our study, we could not differentiate the stromal expression of PD-L1 in immune cells from other stromal cells upon the classification procedure…” what is the potential obstacle regarding this for your study outcome. Please, reflect on this.

- This study focuses on the PD-L1 status in the tumor, stroma and combined stroma + tumor. Have you also looked at other possible candidates as biomarkers for prognosis of ESCC?

Major revision points:

o Figure 1 and figure 2 need to be increased size wise, since the image quality and size of images, letters and numbers make it hard to evaluate.

o Table legends are placed below the tables, but should be placed above.

o Table legend for Table 4 is too scarce, statistical method used?

o Table legend for Table 5 is too scarce, statistical method used?

Minor revision points

o Increase the size of the images for Figure S1, as well

o Look over the heterogeneity how the statistics is presented in the text section

Response:  We sincerely appreciate the thoughtful and constructive feedback, which not only acknowledges the novel perspective of our study but also guides us towards enhancing the clarity and impact of our work. We will address each point you've raised, one by one.

Comments 1: -The analysis is based on formalin-fixed paraffin embedded samples and selected on pathological reports. How many pathologists were involved in the selection and confirmation of stained samples in this procedure?

Response 1: We sincerely appreciate your insightful inquiry regarding the selection and confirmation process of the formalin-fixed paraffin-embedded samples based on pathological reports. Your question highlights an important aspect of our methodology, underscoring the need for clarity in our description of the collaborative effort involved.

In our study, the selection of sample sections was guided by the standard practices of the Department of Diagnostic Pathology at our institution, which includes indicating the deepest tumor region in the pathological reports. This practice was particularly instrumental for our research, as the deepest part of the tumor is often where the most aggressive tumor components are located, providing valuable insights into the tumor microenvironment. Upon consultation with the two pathologists who are co-authors of this study, it was advised that these deepest sections are the most suitable for evaluation due to their use in immunohistochemical staining, including D2-40 and Elastica van Gieson (EVG) stains. This advice was based on the premise that these sections not only represent the tumor effectively but also include areas where the tumor interacts most intensely with its surrounding microenvironment. All authors concurred with this approach, considering that selecting the deepest part of the tumor for our analysis would yield the most informative insights regarding PD-L1 expression and its implications on the tumor biology and its microenvironment, thereby enhancing the significance and relevance of our findings.

This feedback has been incorporated into the manuscript at lines 123-126, where we have revised the text to state: "Formalin-fixed paraffin-embedded blocks from the tumor center were selected based on the standard procedures of the Department of Diagnostic Pathology, focusing on the deepest tumor regions as advised by two pathologists, and sections from these blocks were prepared for immunohistochemical analysis. "

For information regarding the pathologists' confirmation, we have added at lines 135-136: “ The pathologists involved in selection contributed to the confirmation of the stained samples. ”

Comments 2 - The use of this specific anti-PD-L1 antibody (ab205921), is this based on previous testing of anti-PD-L1 antibodies? In some cases, as for the detection of PD-L1 expression in lung cancer, the used anti-PD-L1 clone is of great importance for the outcome in the analysis of the cancer samples.

Response 2: We are truly grateful for your detailed inquiry regarding our choice of the anti-PD-L1 antibody for our study. We acknowledge the importance of this decision in our methodology and appreciate the opportunity to clarify the rationale behind our choice. It is recognized that the choice of anti-PD-L1 clone can significantly influence the outcome of PD-L1 expression analysis, a fact that is particularly pertinent in the context of lung cancer and other malignancies. In the clinical setting for esophageal cancer, the PD-L1 22C3 pharmDx is commonly applied for Combined Positive Score (CPS) diagnostics. Similarly, for the assessment of PD-L1 expression in esophagogastric junction cancer and gastric cancer, the PD-L1 IHC 28-8 pharmDx is known to be utilized. Given the constraints of our research budget and the need to analyze a large number of cases, we were unable to use these clinical grade antibodies for our study. Our project, initiated in 2020, coincided with the clinical adoption of nivolumab as a second-line treatment for esophageal squamous cell carcinoma (ESCC). This development influenced our decision to select the 28-8 clone, which has a deep association with nivolumab, aiming to explore its relevance in the context of ESCC. For our research purposes, we chose the ab205921 antibody, which was one of the most extensively used research-grade reagent available for the 28-8 clone including esophageal cancer studies. This decision was made after careful consideration of both the scientific and practical aspects of our study, including the availability of reagents, budgetary constraints, and the clinical relevance of our research objectives. We believe that our choice of the ab205921 antibody allowed us to conduct a comprehensive and meaningful analysis of PD-L1 expression in ESCC, contributing valuable insights to the field.

This content has been noted in lines 128-130 as follows: " For the primary antibody, we used the research-grade rabbit monoclonal anti-PD-L1 antibody [28-8] (ab205921), which has been documented for use in various cancers, in-cluding esophageal carcinoma, in the literature [21,22]. Primary antibody …”

According to this two references were added in the reference section

21.       Wu, X.; et al. J Immunol Res 2020, 2020, 8884683,

22.       Wang, P. ;  et al J Immunother Cancer 2021, 9,

Comments 3 - “in our study, we could not differentiate the stromal expression of PD-L1in immune cells from other stromal cells upon the classification procedure…” what is the potential obstacle regarding this for your study outcome. Please, reflect on this.

Response 3 -Thank you for your insightful question regarding the challenges we faced in differentiating the stromal expression of PD-L1 in immune cells from other stromal cells within our classification procedure. This limitation presents a potential obstacle in fully understanding the complex dynamics of PD-L1 expression within the tumor immune microenvironment (TIME) of ESCC. While our approach was effective in distinguishing between tumor cells and the broader stromal components based on cell morphology and size, further subdivision within the stromal compartment to differentiate various cell types, including immune cells, presented a significant challenge. Despite employing machine learning techniques for image recognition and classification, the complexity and heterogeneity of stromal components necessitated substantial effort to train the algorithms effectively. We attempted to refine our machine learning model to achieve this differentiation within the stromal compartment. However, despite these efforts, the classification accuracy for distinguishing specific stromal cell types, including immune cells from other stromal cells, did not reach a level that we found satisfactory. This limitation is primarily due to the intricate intermixing of various cell types within the stroma, which complicates the task of creating distinct, recognizable patterns for the machine learning algorithm to identify and classify with high precision. This challenge underscores the current limitations of image analysis and machine learning in the context of highly complex and heterogeneous tissue environments. It highlights the need for further advancements in image analysis technologies and machine learning algorithms that can handle the subtleties of cellular morphology and the dynamic interactions within the TIME more effectively.

In response to this, we have added to the Discussion section at lines 340-345 “In our study, we could not differentiate the stromal expression of PD-L1 in immune cells from other stromal cells upon the classification procedure due to the complexity and heterogeneity of stromal components. It is anticipated that future advancements in image analysis and machine learning will enable the differentiation of subtle features in the TIME, thereby facilitating a better understanding of PD-L1 dynamics in ESCC.”

Comments 4 - This study focuses on the PD-L1 status in the tumor, stroma and combined stroma + tumor. Have you also looked at other possible candidates as biomarkers for prognosis of ESCC?

Response 4 - Thank you for your thoughtful question regarding the exploration of additional biomarkers for the prognosis of ESCC beyond PD-L1 status in tumor, stroma, and stroma + tumor. The inception of our study was significantly influenced by the paradigm shift in ESCC treatment towards the use of immune checkpoint inhibitors (ICIs), which has brought the TIME into the spotlight. This shift has not only highlighted the importance of PD-L1 as a biomarker but also opened avenues for exploring other potential prognostic biomarkers within the TIME. Our research, supported by a grant from the Japan Society for the Promotion of Science, is part of a broader initiative to delve deeper into the TIME of ESCC. While PD-L1 has been our primary focus due to its established role in ICI responses, we are indeed looking into other candidates that could serve as biomarkers for ESCC prognosis. Specifically, our interest extends beyond the mere distinction between "Hot Tumors" and "Cold Tumors" to explore the dual nature within the "Hot" TIME itself, which can potentially promote or inhibit tumor growth. We hypothesize that within this complexity, there exists an optimal inflammatory status, similar to the role of PD-L1, that could significantly enhance prognosis and treatment responsiveness. Currently, as part of our next project, we are examining the expression of T-cell markers such as CD3 and CD8, as well as tumor-associated macrophages (TAMs) markers like CD68 and CD163 in consecutive sections. Preliminary data from these studies are encouraging and suggest that a comprehensive analysis of the TIME could provide valuable insights into ESCC prognosis. Furthermore, our exploration of ESCC delves into the detrimental aspects of inflammation, with a particular emphasis on cachexia and the role of FN14 (TNF Receptor Superfamily Member 12A). This investigation seeks to clarify how FN14 may influence the intricate interplay between ESCC and cachexia, grounded in the understanding that while inflammation can have a dual impact on cancer progression, specific inflammatory markers like FN14 could potentially exacerbate the prognosis. In summary, while PD-L1 remains a critical biomarker in our study, we are actively exploring additional biomarkers that could further elucidate the complex interactions within the TIME and their implications for ESCC prognosis. We believe that these efforts will contribute to a more nuanced understanding of ESCC and potentially lead to the identification of new therapeutic targets.

To reflect this, in lines 383-390 we have updated our manuscript to replace the initial statement "Our research provides a foundation for further investigations into the complex mechanisms of PD-L1 expression in both stromal and tumor cells of ESCC. Recognizing that PD-L1 is part of a broader network within the TIME interactions, future studies should continue to explore additional biomarkers for predicting treatment response and identifying new therapeutic targets. Every advancement in this area contributes to our ongoing efforts to improve patient outcomes in ESCC.” → “Building upon our foundational work in delineating PD-L1 expression dynamics within the stromal and tumor cells of ESCC, future research will delve further into the TIME. This exploration includes assessing T-cell and tumor-associated macrophages markers, along with inflammatory cytokines that potentially exacerbate ESCC prognosis through mechanisms related to cachexia. Each step forward in deciphering the intricate interactions within the TIME not only aids in predicting treatment responses but also strives to identify novel therapeutic targets, moving us closer to improving patient outcomes in ESCC.”

Comments 5 --Major revision points: o Figure 1 and figure 2 need to be increased size wise, since the image quality and size of images, letters and numbers make it hard to evaluate.

o Table legends are placed below the tables, but should be placed above.

o Table legend for Table 4 is too scarce, statistical method used?

o Table legend for Table 5 is too scarce, statistical method used?

Response 5:  We are immensely grateful for your detailed and constructive feedback regarding the presentation of our figures and tables. These changes will undoubtedly improve the quality and impact of our work, and we are thankful for the opportunity to refine our manuscript with your expert guidance.

For all parts of Figures 1 and 2 the images were enlarged and the font size for labels were increased (X-axis, Y-axis, graph labels, data labels, and the number of patients at risk) to the maximum extent.

The table legends were relocated to appear above the tables, as suggested. This change has been applied to Table 1 (Lines 118-119), Table 2 (Line 218), Table 3 (Lines 236-237), Table 4 (Lines 250-251), and Table 5 (Lines 260-262).

Expansion of Content and Inclusion of Statistical Methods: For Table 4 (Lines 250-251), and Table 5 (Lines 260-262), the statistical methods used were added in the table legends in detail to provide clearer insights into our analytical approach.

Comments 6 -Minor revision points

o Increase the size of the images for Figure S1, as well

o Look over the heterogeneity how the statistics is presented in the text section

Response 6 -Thank you so much for your attention to the finer details of our manuscript, specifically regarding the supplementary materials and the presentation of statistical heterogeneity within the text. We are deeply thankful for the time and effort you have invested in helping us enhance the quality of our work.

For all parts of Supplementary Figure S1 (S1A-H), the images were enlarged and the font size for labels were increased (X-axis, Y-axis, graph labels, data labels, and the number of patients at risk) to the maximum extent. For Supplementary Figure S1B, the image was enlarged within the constraints. For Supplementary Figures S1C-H, the font size for labels were increased to enhance the clarity.

To address the issue of statistical heterogeneity, the following sections were revised.

Line 186: The presentation of decimal places in statistical values were standardized (e.g., median 155.0 vs 97.4; P<0.001, Figure 1A), ensuring uniformity across all similar data points.

Line 199: Inconsistencies in the presentation of decimal places for P-values (e.g., P=0.000556) were corrected, aligning them across the manuscript for uniformity.

Line208-216: The description of statistical methods were standardized within figure legends to enhance the manuscript's coherence.

Line 226-227: The notation for confidence intervals (e.g., HR=2.70, 95% CI 1.12-6.52, P=0.027) were standardized to maintain consistency in how these values are presented.

Line 278-291 & Line 304-305: Further modifications were implemented to unify the presentation of statistical values and methods within figure legends and throughout the text (e.g., Median 102.4 vs 106.6, P=0.450; Median 148.5 vs 158.7, P=0.568; Median 108.1 vs 122.3, P=0.287, respectively; Figure 2G-H). 

Line421-425 For Figures S1 and S2, the titles and the statistical methods presented were ensured for consistency with the rest of the manuscript, further contributing to the uniformity and precision of our scientific communication.

Specific adjustments were made to Table 1. Initially, the presentation format of Table 1 resembled a table within a table, which could potentially confuse readers. To address this issue, we have removed the column headings for the description of recurrent cases, simplifying the layout to a single set of column headings.

Your feedback has been invaluable in enhancing the accuracy and readability of our manuscript. We are grateful for the opportunity to improve our work and believe these adjustments significantly contribute to the manuscript's overall quality.

4. Response to Comments on the Quality of English Language

Point 1:

Response 1:    (in red)

5. Additional clarifications

Thank you for the opportunity to provide further clarifications. We have made the following revisions based on the feedback received during the review process:

Line 4, the change from 'Yuuki' → 'Yuki' has been made for the co-author's name, correcting the discrepancy between the Japanese and English spelling as highlighted by the individual.

Line 25 and Line 41 have been updated from 'tumor microenvironment' → 'tumor immune microenvironment' for clarity and to better align with the overall flow of the manuscript."

Line 143 has been revised from 'Department of Pathology' to 'Department of Diagnostic Pathology' from unify the terminology used.

Line 349 has been corrected to spell out 'non-small cell lung cancer' instead of the sudden use of 'NSCLC' without a preceding explanation of the abbreviation.

Reviewer 2 Report

Comments and Suggestions for Authors

Author try to revealed that unique role of PDL1 in esophegal  cancer. So far study have lack of novelty. Proper clinical relevance is missing here. Take message is not clear properly.  

1. What is the main question addressed by the research?   Mentioned in author comment portion.   2. What parts do you consider original or relevant for the field? Current research almost orginal and very relevant for the field.    What specific gap in the field does the paper address? Novel therapeutic strategies.     3. What does it add to the subject area compared with other published material?   This is the review material.     4. What specific improvements should the authors consider regarding the methodology? What further controls should be considered?   This is review material.   5. Please describe how the conclusions are or are not consistent with the evidence and arguments presented. Please also indicate if all main questions posed were addressed and by which specific experiments. 6. Are the references appropriate? Yes 7. Please include any additional comments on the tables and figures and quality of the data.   Not applicable.  

Author Response

March 7, 2024

Dear Reviewer 2,

We are sincerely grateful for your valuable feedback on our manuscript, "Contrasting Roles of PD-L1 Expression in Tumor and Stroma in Prognosis of Esophageal Squamous Cell Carcinoma." Your critique and questions have been crucial in guiding us through the revision process, aiming to enhance the novelty, clarity, and clinical relevance of our study.

In response to your insightful comments, we have undertaken several significant revisions to better articulate the study's contribution to the field, addressing the unique role of PD-L1 in this disease. We believe these revisions significantly strengthen the potential impact of this manuscript. We hope that our efforts now align more closely with the high standards of "Cancers" and that you will find the revised manuscript to be a valuable addition to the field.

Thank you once again for your thorough review and constructive suggestions. We eagerly await your thoughts on the updated version and the opportunity to contribute to this esteemed journal.

Sincerely,

Eisuke Booka, MD, PhD

Second Department of Surgery, Hamamatsu University School of Medicine

1-20-1 Handayama, Chuo-ku, Hamamatsu City, Shizuoka, 431-3192, Japan

Phone: +81-53-435-2279

Response to Reviewer 2 Comments

1. Summary

Thank you very much for taking the time to review this manuscript. Please find the detailed responses below and the corresponding revisions highlighted in the re-submitted files.

2. Questions for General Evaluation

Reviewer’s Evaluation

Response and Revisions

Does the introduction provide sufficient background and include all relevant references?

Yes/Can be improved/Must be improved/Not applicable

Are all the cited references relevant to the research?

Yes/Can be improved/Must be improved/Not applicable

Is the research design appropriate?

Yes/Can be improved/Must be improved/Not applicable

Are the methods adequately described?

Yes/Can be improved/Must be improved/Not applicable

Are the results clearly presented?

Yes/Can be improved/Must be improved/Not applicable

Are the conclusions supported by the results?

Yes/Can be improved/Must be improved/Not applicable

3. Point-by-point response to Comments and Suggestions for Authors

Comments: Author try to revealed that unique role of PDL1 in esophegal cancer. So far study have lack of novelty. Proper clinical relevance is missing here. Take message is not clear properly.

1. What is the main question addressed by the research? Mentioned in author comment portion.

2. What parts do you consider original or relevant for the field? Current research almost orginal and very relevant for the field.  What specific gap in the field does the paper address? Novel therapeutic strategies.    

3. What does it add to the subject area compared with other published material?   This is the review material.    

4. What specific improvements should the authors consider regarding the methodology? What further controls should be considered?   This is review material.  

5. Please describe how the conclusions are or are not consistent with the evidence and arguments presented. Please also indicate if all main questions posed were addressed and by which specific experiments.

6. Are the references appropriate? Yes

7. Please include any additional comments on the tables and figures and quality of the data.   Not applicable. 

Response:  We sincerely appreciate the thoughtful and constructive feedback, which not only acknowledges the novel perspective of our study but also guides us towards enhancing the clarity and impact of our work. We will address each point you've raised, one by one.

Comments 1: What is the main question addressed by the research? Mentioned in author comment portion.

Response 1:  We would like to express our deepest gratitude for the critical assessment of our manuscript which we believe is essential for refining our study. We are committed to making the necessary revisions to address your concerns and to enhance the overall quality and impact of our manuscript. The primary question our study addresses is the comprehensive role of PD-L1 expression in predicting the response to various treatment modalities, including immune checkpoint inhibitors (ICIs) and cytotoxic chemotherapy (CTx), within the context of esophageal squamous cell carcinoma (ESCC). This exploration is motivated by the inconclusive nature of PD-L1 as a predictive marker for ICI response, prompting us to investigate its clinical-pathological significance and impact on prognosis more broadly. Our research is driven by the need to understand the complex interactions between cancer cells, the immune environment, and therapeutic agents in ESCC. We aim to elucidate how PD-L1 expression in both tumor and stromal compartments influences disease progression and treatment outcomes. This involves a nuanced evaluation of PD-L1 expression, considering staining intensity and the differential roles of PD-L1 in tumor versus stromal areas. Our observations have highlighted variability in staining intensity across cases, suggesting that existing evaluation methods, such as the Combined Positive Score (CPS), may not fully capture the significance of staining intensity levels. Additionally, the Tumor Proportion Score (TPS) focuses solely on tumor cells, neglecting stromal expression. By adopting a holistic approach, we hope to shed light on PD-L1's multifaceted role in modulating the immune response and its potential as a biomarker for tailoring treatment strategies in ESCC. This investigation extends beyond the scope of ICIs alone, aiming to provide a clearer understanding of PD-L1's prognostic value and its implications for treatment strategies in ESCC. Our study seeks to contribute to the evolving landscape of ESCC treatment, offering insights that could lead to improved patient outcomes.

In lines 97-109, we have updated our manuscript to replace the initial statement  

“This study investigates the clinicopathological significance of PD-L1 expression in tumor and stromal regions, emphasizing the differential implications between these compartments. Our goal is to understand how PD-L1 expression in ESCC influences disease progression. Such insights could help guide treatment decisions regarding not only ICIs but also CTx in the context of recurrence.”→” This study delves into the clinicopathological significance of PD-L1 expression in ESCC, aiming to comprehensively understand its role in predicting responses to treatment modalities, including ICIs and CTx. Motivated by the ambiguous nature of PD-L1 as a predictive marker for ICI efficacy, our research seeks to broaden the investigation into its impact on disease progression and prognosis. This involves a detailed evaluation of PD-L1's staining intensity and its variable implications in tumor versus stromal areas, aiming to add another layer to current evaluation methods, which may overlook criti-cal aspects of stromal expression. By adopting a holistic approach, our goal is to un-cover the multifaceted role of PD-L1 in reflecting the status of the TIME, and its viabil-ity as a biomarker for customizing treatment strategies in ESCC. This investigation aims to clarify PD-L1's prognostic value and refine treatment paradigms, contributing to enhanced patient outcomes in the evolving landscape of ESCC treatment.”

Comments 2 - What parts do you consider original or relevant for the field? Current research almost orginal and very relevant for the field. What specific gap in the field does the paper address? Novel therapeutic strategies.

Response 2: We sincerely appreciate your recognition of the potential originality and relevance of our work. Your question allows us to highlight the specific contributions our study makes towards further advancing the understanding of ESCC, including therapeutic strategies. Our study addresses a significant gap by applying digital pathology techniques to quantitatively analyze PD-L1 expression in ESCC and potentially other cancers. While the practice of immunohistochemical staining to quantify PD-L1 in tumor tissues is well-established, the full potential of digital pathology for precise reevaluation of PD-L1 has yet to be fully realized. To date, despite the existence of strict guidelines for methods such as the CPS, which are well-established in the clinical setting, the quantification of PD-L1 has largely remained subjective, still relying heavily on the interpretations of pathologists. Despite the emergence of technologies that enable pixel-value-based quantification for a more objective measurement, traditional visual assessments by the human eye, which rely on a 0-3 scale for staining intensity or binary classifications of positive or negative, continue to be valued for their reliability, as evidenced by numerous reports. These approaches do not take full advantage of digital pathology's capability to offer a more detailed and accurate analysis of PD-L1 expression levels, missing the nuances that could be critical for understanding PD-L1's role in the tumor immune microenvironment (TIME). Furthermore, the significance of PD-L1 expression in the stroma, as considered in CPS, is recognized, yet research analyzing stromal PD-L1 expression independently or comparing its clinical-pathological significance with tumor PD-L1 expression in ESCC is scarce. Our literature search found no studies specifically investigating these aspects in the context of ESCC. Additionally, while much of the focus on PD-L1 has been on its predictive value for the efficacy of ICIs, our study explores relatively uncharted territory by examining PD-L1's impact on the effectiveness of CTx in ESCC. This introduces a novel perspective to PD-L1 research.

In lines320-329, we have updated our manuscript to replace the initial statement: “Our approach to evaluating PD-L1 expression in this study represents a significant departure from traditional methods. While prior assessments often relied on subjective, visual scoring systems, our study utilized a more objective H-score-based approach. This method captures variations in staining intensity across the entire slide using OD values for the regions of interest, rather than relying on subjective evaluations by human eye. Enhanced by AI-based tissue classification, this approach was ap-plied to pathologically confirmed tumor regions, as assessed by pathologists on hema-toxylin and eosin stained samples, providing a more precise and standardized meas-urement of PD-L1 expression.”

→”Our approach to evaluating PD-L1 expression in this study marks a significant shift from traditional methods by leveraging digital pathology for quantitative analysis of PD-L1 in ESCC, thus deepening our grasp of its clinical-pathological significance. En-hanced by AI-based tissue classification, this method efficiently differentiates between tumor and stromal regions, spotlighting the underexplored area of stromal PD-L1 ex-pression and its relative clinical-pathological importance in comparison to tumor PD-L1 expression in ESCC. This pioneering approach not only improves the precision of PD-L1 measurement but also expands our insight into its role within the TIME, thereby offering a more nuanced understanding of the interplay between tumor and stromal components in ESCC.”

In lines former text 313-314 we removed “This insight could greatly impact the therapeutic strategy for ESCC, suggesting that evaluating stromal PD-L1 expression could potentially guide treatment decisions.” to avoid repetitive statements, and in lines 359-364 we have updated our manuscript to replace the initial exploration; “This novel insight could expand our understanding of PD-L1's role in treatment efficacy.”→”Our research offers an innovative perspective by examining PD-L1's influence on CTx effectiveness in ESCC, moving beyond the conventional focus on its correlation with ICI responsiveness. By understanding PD-L1 status from resected specimens, we can predict the responsiveness to CTx, which plays a significant role from the first line re-currence treatment, paving the way for more precise and personalized medicine.”

Comments 3 - What does it add to the subject area compared with other published material? This is the review material.

Response 3 - We are deeply grateful for this insightful question, which prompts us to articulate the distinct contributions our study makes to the existing body of knowledge. Your inquiry not only encourages a thorough comparison with published material but also allows us to emphasize the unique value our research adds to the subject area. For many clinicians working with ESCC, the CPS and TPS have been central to evaluating PD-L1 expression. However, the expression of PD-L1 in surgical resection specimens of ESCC presents a wide range of staining intensities and distributions, which are not fully captured by existing evaluation methods. Moreover, these methods do not adequately reflect the significance of PD-L1 expression in both tumor and stromal compartments. Various environmental factors within the TIME that upregulate PD-L1 expression have been reported, indicating a complex interplay at work. Our data add to this area by providing insights into the dynamics of cancer progression related to the upregulation of PD-L1 in both tumor and stromal compartments. The evaluation method we chose, the H-score, although not clinically established like TPS or CPS, distinguishes our work by adequately considering staining intensity. Furthermore, our use of digital pathology to evaluate the entire tumor area within sections marks a significant departure from previously reported studies. By incorporating these novel aspects, our research enriches the field with a more nuanced understanding of PD-L1's role in ESCC. It highlights the importance of considering the full spectrum of PD-L1 expression across the tumor microenvironment, thereby offering a more comprehensive perspective on its implications for cancer progression and potential therapeutic strategies.

This content has been reflected in lines 405-412; “To sum, this research advances the field by elucidating the complex role of PD-L1 in both tumor and stromal compartments of ESCC, employing quantification methods to capture staining intensity nuances not considered by CPS and TPS. The digital pathol-ogy based H-score quantification methods offers a more nuanced view of PD-L1's im-pact on cancer progression and therapeutic avenues. This work underscores the neces-sity of a holistic view of PD-L1 expression within the TIME, contributing significantly to the understanding of its role in ESCC and guiding future therapeutic strategies.

Comments 4 - What specific improvements should the authors consider regarding the methodology? What further controls should be considered? This is review material.  

Response 4 - We are sincerely thankful for your insightful suggestions, which have prompted a deeper reflection on the methodological limitations of our study. Your thoughtful guidance is greatly appreciated as it significantly contributes to strengthening the overall quality of our research. While increasing the sample size could indeed enrich our findings, our current focus leans towards maximizing the insights from our existing dataset. Nevertheless, we agree that conducting a multi-institutional study could significantly bolster the persuasiveness and generalizability of our results. Such a study would benefit from a wider variety of patient demographics and treatment histories, potentially unveiling novel insights into PD-L1 expression and its implications in ESCC.

In terms of diversifying our patient cohorts, the advent of ICIs as a first-line treatment for advanced and recurrent ESCC since our study's inception in 2020 presents a unique opportunity. Although these cases are still relatively few, gathering and analyzing data from such patients could illuminate new research pathways. For instance, investigating PD-L1 expression in patients who have experienced significant tumor reduction and undergone conversion surgery could shed light on mechanisms of resistance to both ICIs and CTx, aiding in the identification of new therapeutic targets.

The integration of multi-omics data is indeed a promising direction for our research. Recognizing the rich potential of genomic, transcriptomic, and proteomic analyses to deepen our understanding of PD-L1 expression and its intricate interactions within the tumor microenvironment, we are exploring avenues to incorporate such data into our studies. This includes leveraging public databases and existing resources, which offer a wealth of information that can be analyzed to uncover the molecular underpinnings of PD-L1 dynamics in ESCC. By harnessing these resources, we aim to overcome the limitations of budget and manpower and to contribute valuable insights into the field.

Longitudinal analysis is indeed critical for understanding the dynamic nature of PD-L1 expression in response to treatment. Future studies designed to monitor changes in PD-L1 levels over time would offer invaluable insights into its prognostic significance and guide more effective treatment planning.

Regarding further controls for digital pathology analysis, we recognize the clinical validation of CPS and TPS. Expanding our dataset to include slides evaluated with these scores, and applying digital image analysis, could provide a meaningful comparison between traditional and digital pathology methods, potentially validating the latter's utility in clinical practice.

Lastly, validation with independent cohorts is crucial for ensuring the reliability and applicability of our findings. Collaborating with other institutions to compile a larger, more diverse dataset could address sample size limitations and lend additional credibility to our conclusions.

This content has been reflected in lines 390-405; “There are several methodological limitations in this study. Future directions will focus on expanding the study scale and diversity by undertaking a multi-institutional ap-proach to compile a larger, more diverse dataset. This expansion should enhance the persuasiveness, generalizability, and credibility of the results by incorporating varied patient demographics and treatment histories. Insights into PD-L1 expression in pa-tients treated with ICIs as a first-line treatment can offer new perspectives on mecha-nisms of resistance and significantly contribute to our understanding of PD-L1’s role in ESCCs. Integration of multi-omics data from public databases and existing resources for genomic, transcriptomic, and proteomic analyses can deepen our understanding of PD-L1 dynamics and interactions within the TIME. Longitudinal analysis will allow us to monitor changes in PD-L1 levels over time, for example, by using endoscopic biopsy specimens before and after treatment could provide insights into PD-L1’s prognostic significance, inform treatment planning, and possibly reveal aspects of treatment re-sponsiveness. Regarding digital pathology analysis, expanding the dataset to include slides evaluated with CPS and TPS will enable a comparison between traditional and digital pathology methods, potentially validating the clinical utility of digital pathology.”

Comments 5 -- Please describe how the conclusions are or are not consistent with the evidence and arguments presented. Please also indicate if all main questions posed were addressed and by which specific experiments.

Response 5- We are deeply grateful for your comprehensive review, which has critically assessed the consistency of our conclusions with the evidence and arguments presented in our manuscript. Your insightful comments have enabled us to more clearly articulate our findings and ensure that our conclusions are robustly supported by the data.

To describe how the conclusions are consistent with the evidence and arguments presented, we incorporated the following statement in lines 364-370 “As discussed so far, the role of PD-L1 in either promoting progression or exerting a suppressive effect on tumors remains controversial in the literature. Our findings, which indicate that high tumoral PD-L1 expression is associated with better prognosis, align with some previous studies in ESCC. Additionally, our report that higher stromal PD-L1 expression is linked to less advanced disease progression and our discovery that PD-L1 expression influences responsiveness to CTx in ESCC both represent firsts in the context of ESCC.”

To indicate if all main questions posed were addressed and by which specific experiments.

In lines 307-320 we have updated our manuscript to replace the initial statement;

“Our study evaluated PD-L1 expression within both the tumor and stromal compartments of ESCC, revealing its complex role in influencing OS and RFS. In the mul-tivariate analysis, tumoral PD-L1 expression emerged as an independent predictor of prognosis. In contrast, high stromal PD-L1 expression was strongly associated with prolonged survival in the Kaplan-Meier analysis and was linked to less advanced pathological stages and improved response to CTx. This discrepancy highlights the potential interplay of multiple confounding factors within the tumor immune micro-environment.”

→ “This study quantified PD-L1 expression in 194 surgical specimens of ESCC using digital pathology, classifying the whole tumor within the slide into tumor and stromal compartments enhanced by AI. By employing Kaplan-Meier survival analysis and categorizing H-scores into quartiles, we found that higher scores were associated with better OS and RFS. Utilizing the minimum P value method for logrank test, we determined statistically optimal H-score cutoff values for OS and RFS, dividing the cohort into groups to assess the impact on survival. In the multivariate analysis regarding survival, tumoral PD-L1 expression emerged as an independent predictor of prognosis for both OS and RFS. Our correlation analysis with clinicopathological variables revealed that high stromal PD-L1 expression was strongly linked to less advanced pathological stages. Furthermore, when comparing H-scores with the duration of CTx and ICI treatment, we discovered that cases treated for more than 8 months exhibited higher stromal H-scores. However, no difference in H-scores between compartments was observed in cases treated with ICI for more than 90 days versus those treated for shorter periods.”

Comments 6 - Are the references appropriate? Yes

Response 6 - Thank you for acknowledging the appropriateness of our references. Careful curation of references was a priority for our team, aiming to accurately reflect the scientific context behind our research and support the novelty and significance of our work.

Comments 7 -Please include any additional comments on the tables and figures and quality of the data.   Not applicable. 

Response 7- Thank you for your comment regarding the tables, figures, and the quality of the data presented in our manuscript. Our team has made concerted efforts to ensure that all visual aids and data presented are of the highest quality and accurately convey the findings of our research.

Your feedback has been invaluable in enhancing the accuracy and readability of our manuscript. We are grateful for the opportunity to improve our work and believe these adjustments significantly contribute to the manuscript's overall quality.

4. Response to Comments on the Quality of English Language

Point 1:

Response 1:    (in red)

5. Additional clarifications

Thank you for the opportunity to provide further clarifications. We have made the following revisions based on the feedback received during the review process:

Line 4, the change from 'Yuuki' → 'Yuki' has been made for the co-author's name, correcting the discrepancy between the Japanese and English spelling as highlighted by the individual.

Line 25 and Line 41 have been updated from 'tumor microenvironment' → 'tumor immune microenvironment' for clarity and to better align with the overall flow of the manuscript."

Line 143 has been revised from 'Department of Pathology' to 'Department of Diagnostic Pathology' from unify the terminology used.

Line 349 has been corrected to spell out 'non-small cell lung cancer' instead of the sudden use of 'NSCLC' without a preceding explanation of the abbreviation.

Reviewer 3 Report

Comments and Suggestions for Authors

First of all, I would like to thank the authors for the work done in writing  the article.

Study assessed PD-L1 expression in both tumor and stromal compartments of ESCC, revealing its complex role in influencing OS and RFS.

The inconclusive results in the literature related to the evaluation of PD-L1 leave the free choice of the scientific collectives to perform various histological evaluations or statistical tests to support their conclusions.

The inclusion of the results in a multicenter study using an identical standardized technique would potentiate an interpretation and an understanding of PD-L1 and PD-1 expression in esophageal cancer.

For a better understanding of the text, the authors should include in the introduction additional explanations for the abbreviations used H-Score and TIME components.

Author Response

March 7, 2024

Dear Reviewer 3,

We appreciate your acknowledgment of our efforts and the constructive feedback provided on our manuscript, "Contrasting Roles of PD-L1 Expression in Tumor and Stroma in Prognosis of Esophageal Squamous Cell Carcinoma." We are particularly thankful for your suggestion to elaborate on the abbreviations used within our manuscript, such as H-Score and TIME components, in the introduction. This advice has guided us to make the text more accessible and informative for all readers, enhancing the manuscript's overall clarity.

We are confident that these revisions have significantly improved the manuscript, making the findings clearer and more comprehensible. We hope that our efforts now meet the esteemed standards of "Cancers" and that you will find the revised manuscript to be a valuable contribution to the field.

Thank you once again for your thorough review and insightful suggestions. We eagerly await your thoughts on the updated manuscript and the opportunity to contribute to this esteemed journal.

Sincerely,

Eisuke Booka, MD, PhD

Second Department of Surgery, Hamamatsu University School of Medicine

1-20-1 Handayama, Chuo-ku, Hamamatsu City, Shizuoka, 431-3192, Japan

Phone: +81-53-435-2279

Response to Reviewer 3 Comments

1. Summary

Thank you very much for taking the time to review this manuscript. Please find the detailed responses below and the corresponding revisions highlighted in the re-submitted files.

2. Questions for General Evaluation

Reviewer’s Evaluation

Response and Revisions

Does the introduction provide sufficient background and include all relevant references?

Yes/Can be improved/Must be improved/Not applicable

Are all the cited references relevant to the research?

Yes/Can be improved/Must be improved/Not applicable

Is the research design appropriate?

Yes/Can be improved/Must be improved/Not applicable

Are the methods adequately described?

Yes/Can be improved/Must be improved/Not applicable

Are the results clearly presented?

Yes/Can be improved/Must be improved/Not applicable

Are the conclusions supported by the results?

Yes/Can be improved/Must be improved/Not applicable

3. Point-by-point response to Comments and Suggestions for Authors

Comments: First of all, I would like to thank the authors for the work done in writing the article. Study assessed PD-L1 expression in both tumor and stromal compartments of ESCC, revealing its complex role in influencing OS and RFS. The inconclusive results in the literature related to the evaluation of PD-L1 leave the free choice of the scientific collectives to perform various histological evaluations or statistical tests to support their conclusions. The inclusion of the results in a multicenter study using an identical standardized technique would potentiate an interpretation and an understanding of PD-L1 and PD-1 expression in esophageal cancer.

For a better understanding of the text, the authors should include in the introduction additional explanations for the abbreviations used H-Score and TIME components.

Response:  We greatly appreciate your positive feedback and thoughtful suggestions on our manuscript. Your insights not only recognize our efforts but also guide us towards enhancing the clarity and impact of our work on PD-L1 expression in ESCC.

For the advise to provide additional explanations for the H-score based quantification technique used in our study. we have expanded our introduction to include a detailed description of the H-score method and its significance in quantifying PD-L1 expression.

This content has been reflected in lines 77-83; “Among the various techniques for quantification, the H-score is particularly effective due to its unique scoring system that emphasizes the presence of strongly stained cells. This method enriches the evaluation of biomarker expression by combining a semi-quantitative assessment of staining intensity with the proportion of positively stained cells. Specifically, it assigns a score from 0 (no staining) to 3 (strong staining) based on the intensity, which is then integrated with the percentage of positive cells to compute the H-score, ranging from 0 to 300 [12] “

Reference 12 by McCarty KS Jr, et al., Cancer Res 1986, 46, 4244s-4248s. was added at the Reference section

For the advise to provide additional explanations for abbreviations (TIME) used in our study.We have also incorporated additional information on the tumor immune microenvironment (TIME) within the same section.

This content has been reflected in lines 83-91→“Recent technological advancements have deepened our understanding of the com-plexity and diversity of the immune context within the tumor microenvironment. It is now recognized that the immune landscape of tumors is not uniform; rather, it com-prises different subclasses of immune environments that significantly influence tumor initiation, progression, and response to therapy. The tumor immune microenvironment (TIME) encompasses a dynamic interplay of various cellular and molecular compo-nents, including immune cells, cytokines, and other factors that can either promote or inhibit tumor growth and metastasis [13].

Reference 13 by Binnewies M, et al., Nat Med 2018, 24, 541-550, doi:10.1038/s41591-018-0014-x. was added at the Reference section

We are grateful for the opportunity to refine our manuscript with your insightful feedback,

believing that these amendments not only address your concerns but also significantly enhance the manuscript's clarity and depth.

4. Response to Comments on the Quality of English Language

Point 1:

Response 1:    (in red)

5. Additional clarifications

Thank you for the opportunity to provide further clarifications. We have made the following revisions based on the feedback received during the review process:

Line 4, the change from 'Yuuki' → 'Yuki' has been made for the co-author's name, correcting the discrepancy between the Japanese and English spelling as highlighted by the individual.

Line 25 and Line 41 have been updated from 'tumor microenvironment' → 'tumor immune microenvironment' for clarity and to better align with the overall flow of the manuscript."

Line 143 has been revised from 'Department of Pathology' to 'Department of Diagnostic Pathology' from unify the terminology used.

Line 349 has been corrected to spell out 'non-small cell lung cancer' instead of the sudden use of 'NSCLC' without a preceding explanation of the abbreviation.

Round 2

Reviewer 2 Report

Comments and Suggestions for Authors

Author clearly indicate the reviewer responses. So now it is acceptable for further publication procedure.